# Peer review of "In Situ Formation of Laser-Cladded Layer on Thin-Walled Tube of Aluminum Alloy in Underwater Environment"

_materials, 2021, doi:10.3390/ma14164729_

Round 1

Reviewer 1 Report

Dear Authors,

The reviewed manuscript titled: “In-situ formation of laser-cladded layer on thin - walled tubular structure of aluminum alloy in underwater environment” describes an experimental study aimed at assessing the weldability issues of underwater local dry laser welding of Al alloy. The subject matter is very important, especially for the shipbuilding and offshore industry. The article deals with a current and relevant issue, it is well prepared, but I have a few comments, which I present below:

  1. Title: I propose to replace: "tubular structure" with "tube"
  2. Abstract: the first three sentences are not well worded, I suggest rewriting them.

Line 18: remove a typo in "underwater".

Keywords: "Thin-walled tubular structure of aluminum alloy" is not a good wording. I think they can be replaced with "Aluminum alloy" and "Thin-walled tube".

  1. Introduction: the section is well prepared. Alternatively, I can recommend adding information that under the conditions of the local dry method, other processes can be used: GMAW, FCAW, TIG.
  2. Chapter 2:

Line 111: add a space before mm

Line 121: Replace: "corrosion liquid" with: "etchant"

All devices used in the tests should be described in accordance with the MDPI rules.

Please enter designation (purity) of Ar.

Lines 146/147: please add a reference.

Chapter 3:

What do the error bars show? Standard deviation?

All figures are too small. I think they can be increased by 10%.

Line 298: Please add a description of the coefficient k. It is less common in the literature.

Line 301: This sentence requires a reference.

Line 313: Correct typo in "that"

Conclusions: I suggest adding an introductory sentence.

Line 327: Correct a typo in "underwater"

References are well chosen and up to date. I suggest replacing [1] with a newer and indexed article by the same author: 10.1515/pomr-2017-0038.

Author Response

Detailed Response to Comments

Honorable reviewers and editors:

Thank you very much for your comments on our paper entitled “In-situ formation of laser-cladded layer on thin - walled tubular structure of aluminum alloy in underwater environment”. Based on your comment and request, we have made extensive modification on the original manuscript. Here, we attached revised manuscript for your approval. A document answering every question from you was also summarized. A revised manuscript with the correction sections yellow highlighted was attached as the supplemental material and for easy check/editing purpose. Here below is our description on revision according to your comments.

(1) Comments of Reviewer #1:

Title: I propose to replace: "tubular structure" with "tube"

Author’s answer:

We agree with this comment and have replaced(2) Comments of Reviewer #1:

Abstract: the first three sentences are not well worded, I suggest rewriting them. Keywords: "Thin-walled tubular structure of aluminum alloy" is not a good wording. I think they can be replaced with "Aluminum alloy" and "Thin-walled tube".

Author’s answer:

We agree with this comment and have already modified.(3) Comments of Reviewer #1:

Introduction: the section is well prepared. Alternatively, I can recommend adding information that under the conditions of the local dry method, other processes can be used: GMAW, FCAW, TIG.

Line 111: add a space before mm

Line 121: Replace: "corrosion liquid" with: "etchant"

All devices used in the tests should be described in accordance with the MDPI rules.

Please enter designation (purity) of Ar.

Lines 146/147: please add a reference.

Author’s answer:

We agree with this comment and have already modified.(4) Comments of Reviewer #1:

Line 298: Please add a description of the coefficient k. It is less common in the literature.

Line 301: This sentence requires a reference.

Line 313: Correct typo in "that"

Conclusions: I suggest adding an introductory sentence.

Line 327: Correct a typo in "underwater"

References are well chosen and up to date. I suggest replacing [1] with a newer and indexed article by the same author: 10.1515/pomr-2017-0038.

Author’s answer:

Definition of equilibrium partition coefficient k: the ratio of solute concentrations in two equilibrium phases of solid and liquid at a certain temperature.We agree with this comment and have already modified.

Reviewer 2 Report

Thank you very much for the opportunity to review this article. The topic is interesting because aluminum should be kept clean during welding and surfacing (cladding/ pad welding), and moisture (water) causes porosity and lower plastic properties. The experiment includes a cladding process in humid/wet conditions (thin water film). I assume that during the process all the water and moisture from the padding point was evaporated, hence the correct shape and structure of the padding weld were obtained.

I have only few notice:

Optical digital microscopy – rather light microscopy

Multiple track cladding layer – rather multiple pass or multiple bead cladding layer

Table 1 – please add separate value for Si and Fe

Line 225-226 – “Due to the influence of molten pool, the microstructure in the HAZ were relatively coarse equiaxed grains.”  - the molten pool dose not influence on crystallization and structure – rather heat input and cooling rate. The heat input and cooling rate and the temperature gradient can influence on microstructure, crystallization, precipitation or just microstructure morphology. The part between line 225 and 231 need be verified and modified.

Fig 6 and Fig 7 need to be changed. The marked notice can be true, but in present form grain boundaries and dendrites are not visible. Please use additionally picture with higher magnification.

The grain size analysis is discussable because you have dendritic microstructure, where the measured size strongly depend on cross-section. How many pictures and at what magnification were analysed for presented figures 9 and 10. Based on the fig. 6, 7 oraz 11 ans 12 such analysis is impossible. Based on fig. 8 and line 267 and 268 may be also subjected to major flaw. In line 123 you only mentioned that the cladding samples size will be analysed – it is necessary to indicate how.

Lines 298-308 are not result of test. It should be removed form paper.

Did you find any Mg-Al eutectic or Al-Mg phase in layer microstructure?

Author Response

Detailed Response to Comments

Honorable reviewers and editors:

Thank you very much for your comments on our paper entitled “In-situ formation of laser-cladded layer on thin - walled tubular structure of aluminum alloy in underwater environment”. Based on your comment and request, we have made extensive modification on the original manuscript. Here, we attached revised manuscript for your approval. A document answering every question from you was also summarized. A revised manuscript with the correction sections yellow highlighted was attached as the supplemental material and for easy check/editing purpose. Here below is our description on revision according to your comments.

(1) Comments of Reviewer #2:

Table 1 – please add separate separate value for Si and Fe

Author’s answer:

The separate value for Si and Fe is a range. We have already modified.
(2) Comments of Reviewer #2:

Line 225-226 – “Due to the influence of molten pool, the microstructure in the HAZ were relatively coarse equiaxed grains.”  - the molten pool dose not influence on crystallization and structure – rather heat input and cooling rate. The heat input and cooling rate and the temperature gradient can influence on microstructure, crystallization, precipitation or just microstructure morphology. The part between line 225 and 231 need be verified and modified.

Author’s answer:

We have made modifications. We referred to the literature: “The rapid loss of heat through the substrate at the edge of the molten pool, resulting in large G and small R, thus forming columnar dendritics in the MZ. Since the direction of temperature gradient was perpendicular to the center of the molten pool from the fusion line, the columnar dendritics grew almost perpendicular to the fusion line toward the center of the molten pool.” The heat input and cooling rate and the temperature gradient can influence on microstructure, crystallization, precipitation. The part between line 225 and 231 have be modified and verified by reference-Investigation on in-situ laser cladding coating of the 304 stainless steel in water environment.

(3) Comments of Reviewer #2:

Fig 6 and Fig 7 need to be changed. The marked notice can be true, but in present form grain boundaries and dendrites are not visible. Please use additionally picture with higher magnification.

Author’s answer:

We agree with this comment and have already modified.(4) Comments of Reviewer #2:The grain size analysis is discussable because you have dendritic microstructure, where the measured size strongly depend on cross-section. How many pictures and at what magnification were analysed for presented figures 9 and 10. Based on the fig. 6, 7 oraz 11 ans 12 such analysis is impossible. Based on fig. 8 and line 267 and 268 may be also subjected to major flaw. In line 123 you only mentioned that the cladding samples size will be analysed – it is necessary to indicate how.Author’s answer:Figures 9 and 10 are based on OZ and MZ of figure 8. We analyzed one photo each of OZ and MZ, and they were about magnified 33 times. Figures 6, 7, 11 and 12 show the dendrites, while figures 9 and 10 show the grain size, and the same dendrite orientation is a grain. Therefore, the idea that based on the fig. 6, 7 oraz 11 ans 12 such analysis is impossible is inextricable. How the cladding is analyzed is described in detail in the following article. We referred to the literature: “To further describe the geometry of the cross section, the thickness of CZ (H, μm), the depth of MZ (h, μm), the width of CZ (W, μm), the width of track band (w, μm) and the angle between adjacent CZ (α, °) were measured, and the dilution rate (D) and the shape factor (S) were evaluated.”(5) Comments of Reviewer #2:

Lines 298-308 are not result of test. It should be removed form paper.

Did you find any Mg-Al eutectic or Al-Mg phase in layer microstructure?

Author’s answer:We think this section is an explanatory note to the above discussion.That is a good suggestion, we will check and modify it.

Reviewer 3 Report

The present manuscript entitled “In-situ formation of laser-cladded layer on thin-walled tubular structure of aluminium alloy in underwater environment” reports a comparative study between two laser-wire cladding processing conditions of aluminium 5052: in air and local dry underwater. More specifically, the authors performed a metallurgical comparison for fixed cladding parameters but in two different environments. Results show a slight difference of morphology and exterior aspect of the cladded track, but more interestingly, a decrease of segregation phenomena and precipitated phases in underwater conditions. It would be very interesting to have a deeper discussion of the results.

The manuscript is overall well written. A consistent use of the abbreviation introduced in the manuscript would ease the reading even more. 

Major considerations:

  • The introduction seems to emphasize a lot the powder-based underwater cladding process that is not used in this research.
  • Is the process fully underwater or local dry underwater process? If so, please state in the introduction, in the methods sections and in the captions of Figure 1 and Figure 2 that this manuscript deals with local dry UWLC technique. Some details are shown in Figure 2 however Figure 1 could be misleading, please revise slightly the schematic on Figure 1; or include Figure 2 into Figure 1. Please add the wire to the Figure 2 so that readers are not misled towards powder-based process.
  • Please state how the process parameters stated in section 2 and in Table 2 were chosen, state eventually preliminary experiments or references. Is the gas flow rate referring to Argon or Air (some pieces of information are not listed). Please define what is considered as ‘rotation angle’.
  • The section 3.1 is introducing the results and discussion section but don’t demonstrate significant results: Formula 1 is stated without further use of it, and would better fit in the introduction. Have the authors had access into the underwater chamber (visual, temperature, particles, etc) ? Are the conclusions stated at the end of section 3.1 corroborated by experiments, hypotheses or facts (i.e. needing references)? Again, Figure 2 would fit better in section 2 and is anyway not referred to in section 3.1.
  • Please discuss further the results.
  • Please give more contextual information of the reported results in the conclusion, such as material of substrate and wire, that the manuscript deals with a comparative study and not an optimization, the motivation of the study, etc.

Minors considerations:

  • In the abstract, there is no need for abbreviations that are not used.
  • Figure 1 describes a ‘UMLC’ nozzle.
  • The abbreviation ‘BM’ appears first in section 3.1 without explanation.
  • The quality of Figure 3a could be improved as it seems blurry.
  • Please check again the grammar of the last paragraph of section 3.1 as some verbs are missing. Please proof read again the sections 3.5 and 4.
  • A great consideration was given to the english but an extensive use of past tense makes the reading somewhat uneasy (e.g. first line of section 3.2).
  • Please add in Figure 4 how you measured the width of the track band (w).
  • Figure 5 could also represent the results of calculating the dilution rate (D) and shape factor (S).
  • The formulas page 5 are not numbered.
  • The section 3.2 compares the cladding in both conditions but the discussion part could be improved.
  • What does “in-air microhardness of material” in section 3.5 mean ?
  • The average microhardness value seems slightly higher for the underwater test.

Author Response

Detailed Response to Comments

Honorable reviewers and editors:

Thank you very much for your comments on our paper entitled “In-situ formation of laser-cladded layer on thin - walled tubular structure of aluminum alloy in underwater environment”. Based on your comment and request, we have made extensive modification on the original manuscript. Here, we attached revised manuscript for your approval. A document answering every question from you was also summarized. A revised manuscript with the correction sections yellow highlighted was attached as the supplemental material and for easy check/editing purpose. Here below is our description on revision according to your comments.

(1) Comments of Reviewer #3:The introduction seems to emphasize a lot the powder-based underwater cladding process that is not used in this research.Is the process fully underwater or local dry underwater process? If so, please state in the introduction, in the methods sections and in the captions of Figure 1 and Figure 2 that this manuscript deals with local dry UWLC technique. Some details are shown in Figure 2 however Figure 1 could be misleading, please revise slightly the schematic on Figure 1; or include Figure 2 into Figure 1. Please add the wire to the Figure 2 so that readers are not misled towards powder-based process.Author’s answer:We add the wire to the Figure 2.(2) Comments of Reviewer #3:Please state how the process parameters stated in section 2 and in Table 2 were chosen, state eventually preliminary experiments or references. Is the gas flow rate referring to Argon or Air (some pieces of information are not listed). Please define what is considered as ‘rotation angle’.Author’s answer:We did do a lot of experiments and explored good welding parameters, and have written an article for publication, but the article is under review and cannot be cited for now. We have already modified the gas flow rate referring to Argon.In addition, a rotary clamping device is designed for clamping the workpiece, and its angle is defined as the rotation angle.(3) Comments of Reviewer #3:The section 3.1 is introducing the results and discussion section but don’t demonstrate significant results: Formula 1 is stated without further use of it, and would better fit in the introduction. Have the authors had access into the underwater chamber (visual, temperature, particles, etc) ? Are the conclusions stated at the end of section 3.1 corroborated by experiments, hypotheses or facts (i.e. needing references)? Again, Figure 2 would fit better in section 2 and is anyway not referred to in section 3.1.Please discuss further the results.Please give more contextual information of the reported results in the conclusion, such as material of substrate and wire, that the manuscript deals with a comparative study and not an optimization, the motivation of the study, etc.
Author’s answer:Based on guo et al. 's researches, we further derive formula 1, which is to illustrate the relationship between laser intensity and aerosol particles. Therefore, we put it in section 3.1. We didn't had access into the underwater chamber and the conclusions stated at the end of section 3.1 corroborated by experiments. We referred to the literature:“As the laser cladding process changed from in-air to underwater environment, the average H decreased from 741 μm to 834 μm, and the average h decreased from 625 μm to 412 μm, besides, based on formula (2), the D decreased from 0.46 to 0.33. The results showed that the peak temperature in the underwater multiple tracks laser cladding process was lower than that of the in-air multiple tracks cladding process under the same process parameters.” Figure 2 has referred to in section 3.1. Since it is a schematic diagram of the laser cladding process, it is put in section 3.1.We've already discussed further the results. We referred to the literature:“By comparing the geometry characteristics of in-air and underwater multiple tracks cladding layers, it was further verified that the existence of water environment could lead to the decrease of laser power density and the instability of cladding process, and the cooling effect of water environment increased the cooling rate of the deposited metal and reduced the high temperature residence time.”We've already added more contextual information of the reported results in the conclusion. We referred to the literature: “In order to study the effect of water environment on wire-feed laser cladding of 5052 aluminum alloy, we compared the quality of in-air and underwater laser cladding under the same process parameters, and the following conclusions are drawn:.”(4) Comments of Reviewer #3:1.In the abstract, there is no need for abbreviations that are not used.2.Figure 1 describes a ‘UMLC’ nozzle.3.The abbreviation ‘BM’ appears first in section 3.1 without explanation.4.The quality of Figure 3a could be improved as it seems blurry.5.Please check again the grammar of the last paragraph of section 3.1 as some verbs are missing. Please proof read again the sections 3.5 and 4.6.A great consideration was given to the english but an extensive use of past tense makes the reading somewhat uneasy (e.g. first line of section 3.2).7.Please add in Figure 4 how you measured the width of the track band (w).8.Figure 5 could also represent the results of calculating the dilution rate (D) and shape factor (S).9.The formulas page 5 are not numbered.10.The section 3.2 compares the cladding in both conditions but the discussion part could be improved.11. does “in-air microhardness of material” in section 3.5 mean? The average microhardness value seems slightly higher for the underwater test.
Author’s answer:1. We will pay attention to it and have modified2. Correct a typo in "UWLC"3. We have corrected this mistake.4. 5.6.7.8.9.10 We have modified it.11. The in-air microhardness of material means the microhardness of welding workpieces in air for comparison with underwater weldments.

Round 2

Reviewer 2 Report

Thank you for your responses and changes in main text. I think the present form is better than previous.

In my opinion, the paper can be published in present form.

Reviewer 3 Report

The authors considered all reviewers’ comments and significantly improved their manuscript. Please check again what is written in red and non english caracters, Line 235-236.